# Dynamic Indices Fail to Predict Fluid Responsiveness in Patients Undergoing One-Lung Ventilation for Thoracoscopic Surgery

**DOI:** 10.3390/jcm10112335

**Published:** 2021-05-27

**Authors:** Kwan-Hoon Choi, Jae-Kwang Shim, Dong-Wook Kim, Chun-Sung Byun, Ji-Hyoung Park

**Affiliations:** 1Department of Anesthesiology and Pain Medicine, Wonju College of Medicine, Yonsei University, Wonju 26426, Korea; bathurst@yonsei.ac.kr (K.-H.C.); ehddnr2004@gmail.com (D.-W.K.); 2Department of Anesthesiology and Pain Medicine, Anesthesia and Pain Research Institute, College of Medicine, Yonsei University, Seoul 03772, Korea; aneshim@yuhs.ac; 3Department of Thoracic and Cardiovascular Surgery, Wonju College of Medicine, Yonsei University, Wonju 26426, Korea; csbyun@yonsei.ac.kr

**Keywords:** fluid responsiveness, pulse pressure variation, stroke volume variation, thoracoscopic surgery

## Abstract

Thoracic surgery using CO_2_ insufflation maintains closed-chest one-lung ventilation (OLV) that may provide the necessary heart–lung interaction for the dynamic indices to predict fluid responsiveness. We studied whether pulse pressure variation (PPV) and stroke volume variation (SVV) can predict fluid responsiveness during thoracoscopic surgery. Forty patients were enrolled in the study. OLV was performed with a tidal volume of 6 mL/kg at a positive end-expiratory pressure of 5 cm H_2_O, while CO_2_ was insufflated to the contralateral side at 8 mm Hg. Patients whose stroke volume index (SVI) increased ≥15% after fluid challenge (7 mL/kg) were defined as fluid responders. The predictive ability of PPV and SVV on fluid responsiveness was investigated using the area under the receiver-operator characteristic curve (AUROC), which was also assessed according to the right or left lateral decubitus position considering the intrathoracic location of the right-sided superior vena cava. AUROCs of PPV and SVV for predicting fluid responsiveness were 0.65 (95% confidence interval 0.47–0.83, *p* = 0.113) and 0.64 (95% confidence interval 0.45–0.82, *p* = 0.147), respectively. The AUROCs of indices did not exhibit any statistical significance according to position. Dynamic indices of preload cannot predict fluid responsiveness during one-lung ventilation with CO_2_ gas insufflation.

## 1. Introduction

Pulse pressure variation (PPV) and stroke volume variation (SVV) are widely used dynamic preload indices for fluid therapy and hemodynamic optimization under positive ventilation of general anesthesia [1]. However, these dynamic indices are dependent on the heart–lung interaction and, thus, their reliability as predictors of fluid responsiveness has been shown to be inconsistent in one-lung ventilation (OLV) for thoracic surgery [2,3,4,5]. In contrast to open thoracotomy, however, thoracoscopic surgery requires CO_2_ insufflation to acquire an optimal surgical view [6]. CO_2_ gas insufflation causes a change in intrathoracic pressure and compliance of the chest wall and lung. Subsequently, increased intrathoracic pressure in both thoracic cavities may exert a physical influence that can interfere with venous return even under OLV with low tidal volume [7], while maintaining closed-chest conditions. These physiologic changes can theoretically provide the necessary heart–lung interaction that may allow the dynamic indices to predict fluid responsiveness, even during OLV with low tidal volume [8]. In addition, considering that the superior vena cava (SVC) is in the right thoracic cavity, there may be differences in the predictive ability of the dynamic indices between the right lateral decubitus (RLD) position and the left lateral decubitus (LLD) position. In theory, right-sided OLV with CO_2_ insufflation at the left thoracic cavity may convey more robust respirophasic variations to the dynamic indices than left-sided OLV. However, no comprehensive evidence yet exists in that regard. Thus, in the present prospective, observational study, we investigated the efficacy of dynamic preload indicators PPV and SVV to predict fluid responsiveness under these circumstances using CO_2_ insufflation at 8 mm Hg.

## 2. Materials and Methods

### 2.1. Study Population

The current trial was approved by the Institutional Review Board of Yonsei University Wonju College of Medicine, Wonju, Korea, and participants were enlisted at https://cris.nih.go.kr on 28 January 2019 (KCT0003575). After acquiring informed consent from each patient, 40 patients with American Society of Anesthesiologist physical status I, II, or III, aged 20–75 years, who had undergone video-assisted thoracic surgery with CO_2_ gas between January 2019 and October 2020, were enrolled. Patients with a history of heart failure, cognitive impairment, recent myocardial infarction, and/or stroke (within 3 months of surgery), and diuretic use before surgery were excluded from the study. Patients with heart disease including arrhythmia, left ventricular ejection fraction of less than 45%, and calculated right ventricular systolic pressure of more than 40 mm Hg by echocardiography were also excluded.

### 2.2. Anesthetic and Procedural Management

All patients fasted from midnight the day before the surgery and received fluids during the fasting period (plasmalyte 120 mL/h). For premedication, patients received an intramuscular injection of midazolam 1 mg before entering the operating room. In the operating room, ECG, non-invasive blood pressure, pulse oximetry (SpO_2_), and bispectral index (BIS) were monitored in all patients. Anesthesia was induced by propofol (1.5–2 mg/kg), rocuronium (0.6 mg/kg), and remifentanil (0.03–0.08 mcg/kg/min), and BIS score was maintained between 40 and 60 through the control of remifentanil and desflurane dosage. A radial arterial catheter was placed on the arm of the opposite surgical site and connected to the FloTrac system (version 4.0; Edwards Lifesciences, Irvine, CA, USA) and the EV 1000 clinical platform (Edwards Lifesciences, Irvine, CA, USA). We continuously monitored stroke volume, stroke volume index (SVI), cardiac output, cardiac index, and SVV through the FloTrac/EV1000 system. PPV was monitored through the Philips IntelliVue MP70 system (Phillips, Suresnes, France). Inspired oxygen fractions started with 1.0 in the case of OLV, and decreased to 0.5 in the case of 94% or more pulse oxygen saturation (sPO_2_). Tidal volume was set at 6 mL/kg of ideal body weight and positive end-expiratory pressure (PEEP) was set at 5 cm H_2_O. The respiratory rate was adjusted within the range of 10 to 15 breaths per minute and end-tidal partial pressure of CO_2_ was maintained within the range of between 35 and 40 mm Hg. Peak inspiratory pressure was maintained below 35 cm H_2_O. When the CO_2_ gas was injected at 8 mm Hg pressure into the thoracic cage, hemodynamic data were recorded (T1). Subsequently, colloid fluid (balanced hydroxyethyl starch 130/0.4) was loaded at a rate of 7 mL/kg [3,4] of the ideal body weight during 20 min, after which hemodynamic data were recorded (T2). Patients whose SVI at T2 was increased by more than 15% from T1 were defined as fluid responders [9,10]. The mean arterial pressure was targeted to be maintained above 60 mmHg. If a hypo-tensive episode occurred, vasopressors were used at the discretion of the anesthesiolo-gist using the ephedrine as the first line agent and phenylephrine as the second lined agent.

### 2.3. Primary Endpoint and Assessment

The primary endpoint of the current study was to investigate the predictive power of PPV and SVV for fluid responsiveness (≥15% SVI) during one-lung ventilation with CO_2_ gas inflation.

### 2.4. Secondary Endpoint

The secondary endpoint of the study was to investigate the predictive power of PPV and SVV for fluid responsiveness (≥15% SVI) according to the site of OLV and CO_2_ insufflation.

### 2.5. Sample Size Calculation

The null hypothesis was that the area under the curve of the receiver-operating characteristic (AUROC) would be 0.5, and the alternative hypothesis was that it would exceed 0.5. If the expected AUROC is 0.8, α = 0.05, and power = 0.90, 28 patients are needed when the responder and non-responder are assumed to be the same. We included 40 people, considering the potential difference in the ratio of responders and non-responders.

### 2.6. Statistical Analysis

All statistical analyses were performed with SPSS 12.0 software (SPSS Inc., Chicago, IL, USA) programs. Intergroup comparisons between fluid responders and non-responders were made using an independent *t*-test, Mann–Whitney U test, or χ^2^ test as appropriate. Categorical variables were analyzed using chi-square or Fisher’s exact tests. A paired *t*-test was performed to compare the variables measured at timepoints T1 and T2. The correlation between the SVV at T1 and the percent change of SVI was analyzed using the Pearson correlation coefficient. The ability of these indices to discriminate responders was determined by the AUROCs of SVV and PPV. A *p* value less than 0.05 was considered statistically significant.

## 3. Results

In total, 40 patients were screened, and all of them were enrolled. There was no dropout among the enrolled patients. There were 17 responders and 23 non-responders. There were no differences in the basic and operative characteristics of the patients, which included operation time, one-lung ventilation duration, and CO_2_ gas infusion time between the two groups (Table 1). Although not statistically significant, the patients in the responder group were older, sicker and had less urine production.

Vasopressor requirements were similar between the responders and non-responders; they were used during or immediately after the anesthetic induction. Vasopressors were not administered during the assessment of fluid responsiveness.

Hemodynamic comparisons before and after volume expansion are shown in Table 2. The SVI was significantly higher in the non-responder group than in the responder group before the fluid challenge. After volume expansion, mean arterial pressure (MAP), cardiac index, SVI, SVV, and PPV showed significant changes from the baseline in both non-responders and responders.

Both SVV and PPV showed a significant correlation with the percent changes in the SVI before and after the fluid challenge (Table 3). However, the AUROC of SVV and PPV on fluid responsiveness was 0.64 (*p* = 0.147, 95% confidence interval (CI) 0.45–0.82) and 0.65 (*p* = 0.113, 95% CI 0.47–0.83), respectively. Moreover, subgroup analysis of the AUROC of PPV and SVV on fluid responsiveness according to the RLD or LLD position revealed no significant differences according to the site of OLV and CO_2_ insufflation, with none of the variables showing any significant predictive ability at any position (Figure 1).

## 4. Discussion

The present study results suggest that PPV and SVV obtained from the radial artery do not predict fluid responsiveness when OLV (6 mL/kg) was performed with CO_2_ insufflation at 8 mm Hg in the contralateral thoracic cavity. Moreover, these indices could not predict fluid responsiveness regardless of the lateral decubitus position. Although the dynamic indices could not predict fluid responsiveness, both PPV and SVV showed statistically significant correlations with the changes in SVI, while the r values were too small to imply any clinical significance.

The dynamic indices are the observation of the relationship between the difference in stroke volume according to the respiratory cycle during mechanical ventilation and the position in the Frank–Starling curve [11]. PPV and SVV are used as representative dynamic indices, and there are many studies on their application in various clinical conditions [12,13,14,15]. However, few studies have examined the predictability of fluid responsiveness in thoracic surgery, which interferes with the generation of sufficient heart–lung interaction required for the dynamic preload indices. Undoubtedly, optimizing fluid therapy exerts significant influence on outcomes after thoracic surgery [16], as volume-induced lung-injury-released inflammatory cytokines can damage the endothelial glycocalyx layer and increase capillary leakage [17]. In a previous study, OLV with a tidal volume of 8 mL/kg and 25% SVI was used as a criterion for fluid responsiveness in surgery under thoracotomy and showed that the AUROC was 0.90 (95% confidence interval, CI 0.809–0.991) [14]. In another study, OLV with a tidal volume of 5–6 mL/kg and 10% SVI was used as a criterion for fluid responsiveness in thoracic surgery, and the AUROC was only 0.63 (95% CI 0.52–0.74) [3]. These inconsistent results imply different predictive abilities of the dynamic indices to predict fluid responsiveness during OLV depending on the tidal volume, thoracic cavity opening, and the criteria that differentiate responders and non-responders.

Positive insufflation pressure exceeding 5 mm Hg causes the cardiac index, mean arterial pressure, and left ventricular stroke work index to decrease while increasing pulmonary artery and central venous pressure [7]. In addition, continuous CO_2_ insufflation reduces lung and chest wall compliance [18]. Reduced chest wall compliance may alter the magnitude of PPV and SVV, independent of preload [8].

In the present study, when the intrathoracic pressure was increased by insufflating 8 mm Hg of CO_2_ gas, the AUROC of PPV was 0.65 (*p* = 0.112, 95% CI 0.47–0.83) and 0.64 (*p* = 0.147, 95% CI 0.45–0.82), respectively, discouraging their use as a preload index. Several factors may have contributed to the inability of these dynamic indices to predict fluid responsiveness during OLV despite maintaining closed chest conditions and positive intrathoracic pressure in both thoracic cavities. First, a low tidal volume (6 mL/kg) is a widely known major factor that may have contributed to predictive failure [19]. It was thought that the continuously increased intrathoracic pressure by CO_2_ insufflation at 8 mm Hg while maintaining a closed-chest condition would accentuate the small effect of low tidal volume on the circulatory system and, thus, provide the necessary heart–lung interaction. However, our observation proved otherwise. Second, the amount of non-ventilated lung shunt and hypoxic vasoconstriction may have influenced the dynamic indices regardless of the preload of patients [20]. Additionally, the non-ventilated lung did not provide any cyclic changes in intra-thoracic pressure. The effect of hypoxic vasoconstriction and amount of shunt on change of cardiac output is unclear [14]. Third, the heart–lung interaction imposed on the respirophasic changes in the dynamic indices to discriminate patients on the steep portion of the Frank-Starling curve not only relies on changes in venous return, but also on changes in right ventricular afterload and left ventricular transmural pressure, and a transient increase in left ventricular filling [21], which may not be consistent during OLV regardless of the closed-chest condition and CO_2_ insufflation. In contrast to our hypothesis, the present results are consistent with the conclusion that the dynamic indices do not predict fluid responsiveness in thoracoscopic surgery requiring OLV [2,3,20].

Of note, the current study is unique in that we performed secondary analysis regarding the predictive ability of PPV and SVV depending on the decubitus position. Unlike the inferior vena cava, the SVC is located in the right thoracic cavity and the venous return through the SVC is more influenced by respirophasic changes in the intrathoracic pressure compared to that through the inferior vena cava [22]. Thus, we additionally hypothesized that OLV to the right lung while insufflating the left thoracic cavity with CO_2_ (RLD position) would provide a more robust respirophasic change in venous return, and thus in preload, than when the surgery is performed on the LLD position. However, both PPV and SVV showed no predictive ability on fluid responsiveness irrespective of the direction of the lateral decubitus position, which may also be attributable to the abovementioned plausible causes.

This study has the following limitations. First, only lung protective ventilation, 6 mL/kg of tidal volume, was implemented. Therefore, the effect according to the tidal volume could not be confirmed. There is no accepted guideline of tidal volume in one-lung ventilation, but high tidal volume stretches the lung, which increases the risk of acute lung injury and acute respiratory distress syndrome [23]. As such, this degree of tidal volume seems to be the most appropriate and widely used in order to assess actual clinical effectiveness. Second, the SVI was measured by the arterial wave form contour analysis. The gold standard of measuring the cardiac output uses the pulmonary artery catheter, although the pulse contour measurement has also been proven to be reliable [24]. The uncalibrated pulse wave contour analysis method used in the current study provides autocalibration every 20 s according to the patients’ and waveform characteristics, which has also been shown to provide clinical accuracy in cardiaothoracic surgery [25]. Third, this study was not sufficiently powered to address subgroup analyses regarding the secondary endpoints. Fourth, data regarding the influence of OLV and CO_2_ insufflation on PPV, SVV, and SVI would have been informative, but we could not obtain them separately as OLV and CO_2_ insufflation was performed simultaneously. Finally, the predictive power of the dynamic indices could be affected by the definition of the fluid responders and the amount of fluid challenge. In contrast to previous studies using smaller volumes of fluid challenge, we administered 7mL/kg of fluid while using a ≥15% increase in SVI as a cut-off for fluid responsiveness to ensure sufficient preload augmentation, which would minimize false-negative values.

## 5. Conclusions

In conclusion, SVV and PPV could not predict fluid responsiveness in patients undergoing thoracoscopic surgery using one-lung ventilation (6 mL/kg) with CO_2_ insufflation (8 mm Hg) while maintaining closed-chest condition. Even when accounting for the intra-thoracic location of the SVC, these indices could not predict fluid responsiveness whether the patients were in RLD or LLD position.

## Figures and Tables

**Figure 1 jcm-10-02335-f001:**
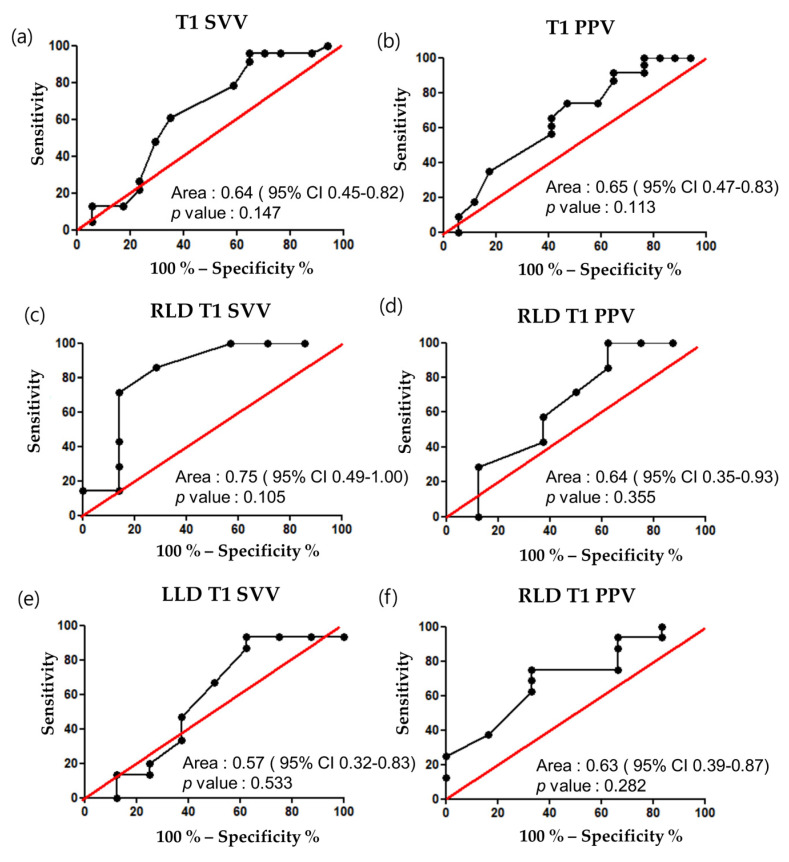
AUROC of dynamic indices before fluid challenge. (**a**) SVV; (**b**) PPV; (**c**) Right lateral decubitus SVV; (**d**) Right lateral decubitus PPV; (**e**) Left lateral decubitus SVV; (**f**) Left lateral decubitus PPV.

**Table 1 jcm-10-02335-t001:** Preoperative characteristics and intraoperative treatment.

Characteristics	Fluid Responders (*n* = 17)	Fluid Non-Responders (*n* = 23)	*p*-Value
Age (years)	67.06 ± 8.44	61.09 ± 12.12	0.090
Male/female	11 (64.7)/6 (35.3)	17 (73.9)/6 (26.1)	
Hypertension	15 (65.2)	6 (35.3)	0.064
Beta blocker	3 (17.6)	1 (4.3)	0.481
ARB/ACEi	8 (47.1)	7 (30.4)	0.386
CCB	6 (35.3)	9 (39.1)	0.850
Diabetes Mellitus	6 (26.1)	3 (17.6)	0.533
Cerebral vascular disease	2 (8.7)	0 (0)	0.218
Body surface area (m^2^)	1.65 ± 0.15	1.75 ± 0.21	0.104
Ideal body weight (kg)	62.47 ± 9.21	67.48 ± 13.67	0.200
ASA (I/II/III)	0/5/12 (0/29.4/70.6)	0/14/9 (0/60.9/39.1)	0.052
HES 130/0.4 infusion (mL)	437.65 ± 58.37	463.91 ± 85.53	0.256
Urine output (mL)	192.83 ± 189.16	355.00 ± 752.84	0.326
Bleeding output (mL)	93.75 ± 134.01	65.22 ± 129.19	0.508
OP time (min)	111.06 ± 72.13	107.87 ± 50.81	0.870
OLV duration (min)	101.82 ± 57.07	105.26 ± 51.41	0.843
CO_2_ time (min)	53.76 ± 43.12	51.30 ± 33.10	0.839
LLD/RLD	9 (52.9)/8 (47.1)	16 (69.6)/7 (30.4)	
Vasopressor			
Ephedrine	11 (64.7)	15 (65.2)	0.974
Dosage (mg)	9.29 ± 9.05	8.96 ± 8.02	0.901
Phenylephrine	1 (5.9)	4 (17.4)	0.283

Data are displayed as mean ± SD, or *n* (%). Abbreviations: HES = hydroxyethyl starch.

**Table 2 jcm-10-02335-t002:** Hemodynamic change in fluid challenge.

	Fluid Responders (*n* = 17)	Fluid Non-Responders (*n* = 23)	*p*1-Value
Heart rate			
T1	76.76 ± 13.11	70.09 ± 14.25	0.138
T2	72.53 ± 11.29	76.70 ± 13.72	0.314
*p*2-value	0.076	0.009	
Mean arterial pressure			
T1	93.18 ± 14.18	87.61 ± 14.07	0.225
T2	76.53 ± 11.01	76.70 ± 10.10	0.961
*p*2-value	0.002	0.002	
Cardiac index			
T1	2.45 ± 0.58	2.74 ± 0.78	0.198
T2	3.02 ± 0.70	3.06 ± 0.72	0.865
*p*2-value	<0.001	0.001	
Stroke volume index			
T1	31.41 ± 8.23	39.78 ± 9.99	0.008
T2	42.35 ± 7.62	40.74 ± 9.66	0.572
*p*2-value	<0.001	0.359	
Stroke volume variation			
T1	10.82 ± 5.58	8.61 ± 3.50	0.144
T2	6.18 ± 2.38	6.22 ± 2.38	0.959
*p*2-value	0.001	0.006	
Pulse pressure variation			
T1	11.06 ± 5.64	8.26 ± 3.67	0.110
T2	5.71 ± 2.44	5.48 ± 2.73	0.485
*p*2-value	<0.001	0.001	
Peak airway pressure			
Gas off	23.91 ± 4.24	25.12 ± 5.01	0.416
Gas on	30.17 ± 4.20	29.53 ± 3.32	0.604
*p*2-value	<0.001	<0.001	

Data are displayed as mean ± SD, T1 is before fluid challenge, T2 is after fluid challenge.

**Table 3 jcm-10-02335-t003:** Relationship between indices and percent change in SVI.

	r	*p*-Value
T1 PPV	0.608	<0.001
T1 SVV	0.553	<0.001

PPV, pulse pressure variation; SVV, stroke volume variation.

## Data Availability

The data presented in this study are available on request from the corresponding author.

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
