# Peer review of "Dynamic Indices Fail to Predict Fluid Responsiveness in Patients Undergoing One-Lung Ventilation for Thoracoscopic Surgery"

_jcm, 2021, doi:10.3390/jcm10112335_

Round 1

Reviewer 1 Report

This is a clean and nice manuscript to read on a well known topic with important literature on it

The paper is well written too

My only comment will be:

1) The volume of the fluid challenge is higher than in the literature....usually 250 ml ( see Messina et al; A&A: Fluid challenge in anesthesia) or even better 4 ml/kg ( see cecconi et al; CCM). 

However, despite that you gave more than a normal fluid challenge, the response was negative which is finally good to confirm that dynamic indices cannot predict fluid responsiveness. Please discuss this point as i explained to you....it is a limitation but finally also a strenght . If the patient does not respond to a higher volume of fluid challenge than recommended, it can really mean that there was not effect.

Author Response

1) The volume of the fluid challenge is higher than in the literature....usually 250 ml ( see Messina et al; A&A: Fluid challenge in anesthesia) or even better 4 ml/kg ( see cecconi et al; CCM). 

However, despite that you gave more than a normal fluid challenge, the response was negative which is finally good to confirm that dynamic indices cannot predict fluid responsiveness. Please discuss this point as i explained to you....it is a limitation but finally also a strenght. If the patient does not respond to a higher volume of fluid challenge than recommended, it can really mean that there was not effect.

Answer: We agree with your comment and have added a sentence to the limitation section concerning your comment as follows: “Finally, the predictive power of the dynamic indices could be affected by the definition of the fluid responders and the amount of fluid challenge. In contrast to previous studies using smaller volumes of fluid challenge, we administered 7ml/kg of fluid while using a ³15% increase in SVI as a cut-off for fluid responsiveness to ensure that sufficient preload augmentation was done, which would minimize false-negative values.”

Thank you for your valuable comments.

Reviewer 2 Report

Dear authors, thank you for considering me as a potential reviewer. This is quite an interesting study as adequate fluid substitution is still challenging especially in thoracic surgery. However, there are some issues that have to be adressed within a major revision:

1) In-/exclusion critiera: all patients presented with sinus rhythm? Was AFib an exclusion criterium? In addition, was left and right ventricular function comparable preoperatively (echo data)?

2) Was the power analysis adequate? The reviewer would doubt this as the authors more or less compare 4 groups: responders vs. non-responders, LLD vs. RLD! In addition, all results have to be separated into these groups (e.g. can you compare a responsive patients in LLD with a responsive patient in RLD and viceversa) - Please comment

3) On what basis fluid substitution was performed with 7ml/kg colloid solution? What was the weight of included patients? The reviewer would guess that most patients received ≥ 500ml as bolus. In addition, over which time period was the bolus infused?

4) Aiming at an SVI increase > 15% seems to be very high. Most other studies declare the cut-off ≥ 10-11%.

5) Is an uncalibrated pulse contour analysis validated in thoracic surgery at all? Why did the authors not use an calibrated pulse contour analysis?

6) When looking a Table 1 the reader gets the impression that patients in the responder group (though not statiscally significant) were older, sicker and had a far less urine production. - Please comment. In this context the authors have to present more details on the preop status and medication (e.g. b-blocker use). - Please comment: are hypertensive patients more prone to fluid responsiveness? In addition, can the authors be sure that preoperative fasting was comparable between both groups? Why didn´t the authors measure pre induction SVI? Is there really no statically significant difference in urine output?

7) Was there a protocol to maintain MAD ± Vasopressors? What information can the reader draw form the given MAD values before/after fluid substitution. Was fluid responsiveness triggered/masked by inadequate vasopressor infusion? In this context it is of interest to also display dosages of used vasopressors at both time points.

8) In the non-responder group there is no SVI increase (Table 2). However, nearly the same SVI in this group is declared as being highly significant different between both time points (p=0,006). Whereas "comparable" SVI values between both groups after fluid substitution are insignificant. - Please comment on this discrepancy

9) Finally, though SVI did not "increase by > 15%", the authors describe a significant decrease in PPV and SVV in the non-responder group. Unfortunately, this observation was not disucussed at all. In addition, it would have been of benefit if the impact of CO2 insufflation on SVI as well as induction of anesthesia was displayed within the results.

At the end, it is not quite clear to the reviewer if the methods were adequate to answer the scientfic question as well as the validity of the here presented results.
